# Curcumin as a Natural Approach of Periodontal Adjunctive Treatment and Its Immunological Implications: A Narrative Review

**DOI:** 10.3390/pharmaceutics14050982

**Published:** 2022-05-03

**Authors:** Sorina Mihaela Solomon, Celina Silvia Stafie, Irina-Georgeta Sufaru, Silvia Teslaru, Cristina Mihaela Ghiciuc, Florin Dumitru Petrariu, Oana Tanculescu

**Affiliations:** 1Department of Periodontology, Grigore T. Popa University of Medicine and Pharmacy Iasi, Universitatii Street 16, 700115 Iasi, Romania; sorina.solomon@umfiasi.ro (S.M.S.); silvia.teslaru@umfiasi.ro (S.T.); 2Department of Preventive Medicine and Interdisciplinarity, Grigore T. Popa University of Medicine and Pharmacy Iasi, Universitatii Street 16, 700115 Iasi, Romania; celina.stafie@umfiasi.ro (C.S.S.); florin.petrariu@umfiasi.ro (F.D.P.); 3Department of Morpho-Functional Sciences II—Pharmacology and Clinical Pharmacology, Grigore T. Popa University of Medicine and Pharmacy Iasi, Universitatii Street 16, 700115 Iasi, Romania; cristina.ghiciuc@umfiasi.ro; 4Department of Fixed Prosthesis, Grigore T. Popa University of Medicine and Pharmacy Iasi, Universitatii Street 16, 700115 Iasi, Romania; oana.tanculescu@umfiasi.ro

**Keywords:** curcuma, local periodontal therapy, oral rinses, gels, photodynamic therapy, photosensitizer, biotolerance, allergy

## Abstract

Scaling and root planing represent the gold standard in the treatment of periodontal disease, but these therapeutic methods cannot eliminate the remaining periodontopathogenic bacteria in cement, tubules, and periodontal soft tissue. Thus, a number of additional therapeutic means have been adopted, including local and systemic antibiotic therapy, as well as the use of photodynamic therapy techniques. Recently, special attention has been paid to potential phytotherapeutic means in the treatment of periodontal disease. In this review, we aim to present the effects generated by the extract of *Curcuma longa*, the various forms of application of turmeric as an additional therapeutic means, as well as the aspects related to its biotolerance.

## 1. Introduction

Periodontal disease is a multifactorial inflammatory condition of the tissues supporting and maintaining the functionality of teeth on the dental arches. The onset of periodontitis is triggered by dysbiosis in the microorganisms of the periodontal biofilm [1,2], to which the host will react through nonspecific and specific defense systems, generating a cascade of inflammatory reactions, in which oxidative stress also plays an important role in sustaining the disease [3]. Among the main periodontal pathogens incriminated in the onset and evolution of periodontal destruction are bacteria from the red complex of Socransky [4], *Porphyromonas gingivalis*, *Tannerella forsythia,* and *Treponema denticola*, but also highly aggressive bacteria, such as *Aggregatibacter actinomycetemcomitans*. Periodontitis proceeds in apical directions [5], involving deeper tissues and causing bone destruction that can lead to tooth loss [6].

Significant data on the strong relationship between periodontitis and systemic diseases have emerged in the last decades [7,8,9,10], and the concept of “periodontal medicine” takes an important role in the complex, interdisciplinary approach to treating a patient with periodontitis.

Periodontal therapy has long been focused on disrupting and eliminating the pathogenic biofilm [11], but the mechanisms involved in the pathophysiology of periodontitis are multiple, recognizing local and systemic risk factors that can negatively affect the quantity and quality of the bacterial plaque but also the immune response of the host [12]. Thus, current therapies for periodontal disease involve, in addition to scaling and root planing (SRP), which remains the gold standard in periodontal treatment, additional therapeutic means, methods that come to the aid of patients with severe periodontitis forms, which are difficult to control [13].

Given that classical mechanical debridement cannot generate a complete removal of periodontal pathogens, antiseptics and antibiotics, with systemic and/or local administration forms, have been adopted as a therapeutic method [14,15]. Of course, the systemic intake of antibiotics brings with it a number of disadvantages, such as hepatic and renal toxicity, the emergence of resistant microorganisms, and the need for high doses in order to achieve an adequate concentration in periodontal tissues [13]. Therefore, local antibiotics and antiseptics may represent an effective alternative against periodontal pathogens, with fewer adverse effects [16]. Such methods include oral rinsing, the use of gels at home, by patients, or professional methods of in situ delivery of antimicrobial agents in the form of instillations, gels, fibers, chips, films, microparticles, nanoparticles, antimicrobial photodisinfection techniques for periodontal pockets, or laser or LED technologies [17].

Various active substances have been investigated, including chlorhexidine, tetracyclines, and metronidazole, which in turn have side effects after long-term administration, including the development of dysbiosis, allergies, and teeth and mucous membranes staining [17]. Therefore, phytotherapeutics have emerged as a relatively cost-effective alternative, with multidirectional efficacy and fewer side effects. Phytotherapeutics with applications in dentistry include aloe vera, cinnamon, neem, cloves, and turmeric [18]. The aim of this review is to revise the main properties of curcumin, including those related to its tolerance by the human body, as well as its applications in adjunctive periodontal therapy.

## 2. Properties of Curcumin

Turmeric is a rhizome of *Curcuma longa*, a member of the ginger family; it is a yellow, water-soluble pigment derived from perennials belonging to the Zingiberaceae family [19]. Curcumin [1,7-bis (hydroxyl-3-methoxyphenyl)—1,6-heptadiene—3,5-dione] is the main turmeric product responsible for biological effects, isolated for the first time in 1815 [20].

Curcumin is a stable product at low pH in aqueous alcohol solutions; it undergoes hydrolysis and chemical degradation at a basic pH [21]. The degradation is significant, even under physiological pH conditions, in the presence of isotonic phosphate buffer. Curcumin follows degradation processes in both solution and solid form [21]. The photodegradation of curcumin generates the accumulation of photoproducts in aerated and deaerated organic solutions and aqueous micellar solutions [22]. Curcumin is subjected to UV photolysis in organic solvents and exposure to sunlight generates more degradation products than UV photolysis [23].

The uses of curcumin worldwide are multiple, from medical compounds to nutritional, cosmetic or industrial uses [24]. As a product, curcumin can be found in solutions, powders, tablets or ointments. Curcuminoids, components of turmeric, are polyphenols, represented by curcumin (diferuloyl-methane) (80%), demethoxycurcumin (15%), and bisdemethoxycurcumin (5%) [25], compounds approved by the US Food and Drug Administration (FDA) as “General Recognized as Safe” (GRAS) [26]. Curcumin is nontoxic; about 40–85% of the oral intake of curcumin passes unaltered through the gastrointestinal tract, with most of the flavonoids consumed being used in the intestinal mucosa and liver [27]. Clinical trials have not yet identified a maximum tolerated dose of curcumin in humans; after administration of up to 8000 mg/day of curcumin, it was concluded that curcumin is nontoxic and has minimal adverse effects on the human body [28]. Due to its low rate of ingestion, curcumin is regularly accompanied by bromelain for extended retention and an enhanced calming effect [29].

Curcumin has been shown in numerous in vitro and in vivo studies to have anti-inflammatory, antioxidant, anticancer, and chemopreventive properties [30,31,32,33] (Figure 1). Curcumin has been shown to decrease markers of oxidative stress and increase antioxidant activity by modulating superoxide dismutase, catalase, or glutathione peroxidase [34,35,36,37]. Inhibition of the lipo-oxygenase/cyclooxygenase and xanthine hydrogenase/oxidase enzymes has also been observed [32,38].

Inflammatory phenomena, with the release of proinflammatory molecules, are involved in the production of oxidative stress mediators; moreover, these products can, in turn, trigger an intracellular signaling cascade, which leads to increased expression of proinflammatory genes [39]. Curcumin has been shown to inhibit the activation of nuclear factor NF-κB, a molecule that stimulates a number of inflammatory markers, such as TNF-α [40,41]. Other studies suggest that the anti-inflammatory mechanism may be due to blocking the metabolism of arachidonic acid, with the following phenomena: selective inhibition of prostaglandin E2 and thromboxane synthesis, inhibition of arachidonic acid metabolism by lipoxygenase, elimination of the generated free radicals, and down-regulation of inflammatory cytokine expression IL-1β and IL-6 [42,43]. Curcumin also inhibits the release and regulation of several matrix metalloproteinases (MMPs) and reduces the release of many proteolytic enzymes, such as elastase, collagenase, and hyaluronidase, from activated macrophages [44,45].

Curcuminoids modulate gene expression and the activity of enzymes involved in lipoprotein metabolism, with a decrease in plasma triglycerides and cholesterol [46], as well as an increase in HDL-C [47]. Moreover, a number of studies demonstrated the induction of cellular apoptosis by curcumin, as well as its antiangiogenic action, with applications in cancer treatment [48,49,50,51].

Given these properties, the potentially beneficial role of curcumin has been investigated in various systemic conditions and diseases, such as osteoarthritis [52], diabetes [53], cardiovascular disease [54,55], metabolic syndrome [56,57], various forms of cancer [58,59], and in oral cavities for the treatment of oral lesions [60] (Figure 2).

We aim to present a series of data from the literature on different methods of local periodontal delivery of curcumin, along with biotolerance and allergologic concerns regarding the curcumin topical usage.

## 3. Oral Rinsing and Irrigation

There are many studies in the literature that support the antibacterial effect of turmeric mouthwash [61]. Mohammed et al. [62] demonstrated the antimicrobial activity of curcuma and Waghmare et al. [63] reported a significant reduction in total microbial count following oral rinsing with curcuma mouthwash.

A number of studies have investigated the effects of turmeric in oral rinsing solutions, supporting its beneficial effects on periodontal clinical parameters, but also on bacterial load and inflammatory and oxidative-stress molecular changes [64]. Most studies have used chlorhexidine as a positive control element, its beneficial effects being clearly stated and demonstrated in the literature [65]. Arunachalam et al. [42] evaluated and compared the effect of curcumin mouthwash with chlorhexidine mouthwash and saline rinses on clinical parameters and salivary levels of reactive oxygen metabolites in patients with chronic gingivitis, as an adjunct to SRP. The parameters were evaluated at baseline and at the end of four weeks, with significant improvements for all groups. However, the authors found a more significant decrease in reactive oxygen metabolites for patients who rinsed with curcumin mouthwash [42]. Higher clinical anti-inflammatory effects of rinses with 1% curcumin solution compared to chlorhexidine were also observed by Suhag et al. [66], with a reduction in periodontal inflammatory edema.

Muglikar et al. [67] compared the effects on the gingival index and plaque index of oral rinses with chlorhexidine and curcumin as adjuvants for SRP on a weekly basis for three weeks, observing similar effects of these two substances, which were more beneficial than SRP alone. Chatterjee et al. [68] also obtained similar results for curcumin versus chlorhexidine oral rinses, in terms of gingival bleeding, plaque index, and gingival index; curcumin was well tolerated, biocompatible, and acceptable in taste [68]. A study by Chainani-Wu [69] showed that curcumin, in addition to its mechanical therapeutic strategies, can be used as a complementary therapy to reduce inflammation; poorer results were observed for the plaque index. Two studies by Gottumukkala et al. [70] and Jalaluddin et al. [71] reported that 0.2% chlorhexidine mouthwash had greater effects on clinical parameters, such as plaque index, gingival index, and bleeding-on-probing index, than mouthwashes with 1% curcumin.

## 4. Gels

A number of studies have investigated the potential beneficial effects of topical application of various turmeric gel formulations in periodontal pockets. Dave et al. [72] investigated the effects of turmeric gel as adjunctive therapy to SRP in periodontitis patients, resulting in significant reductions in plaque index, bleeding on probing, probing depth, and periodontal clinical attachment loss [72]. Jaswal et al. [73] compared the effects on probing depth and clinical attachment loss following a single application of 2% whole turmeric gel to 1% chlorhexidine gel in patients with chronic periodontitis; the obtained results showed significant reductions in periodontal parameters following the application of turmeric gel [73]. Kandwal et al. [74] investigated the effects of applying turmeric gel or chlorhexidine gel for 21 days, showing significant reductions in plaque and gingival index, similar for both substances; in addition, turmeric gel was more easily tolerated by patients than chlorhexidine gel, which caused a bitter taste sensation and dental pigmentation [74]. Similar clinical results were obtained by Anuradha et al. [14] and by Hugar et al. [75] in interventional studies with split-mouth design, which evaluated the effects of turmeric gel application, compared to SRP alone [14] or chlorhexidine gel [75] applied in deep periodontal pockets. In addition, the curcuma gel generated more significant improvements in the periodontal parameters when compared to the ornidazole gel [76]. Nasra et al. [77] developed a gel composed of a mixture of F127 pluronic and P934 carbopol, demonstrating that it could significantly reduce probing depth, improve bleeding index, and inhibit plaque growth as an adjunct to SRP. In contrast, another study did not observe statistically significant differences between chlorhexidine gel and curcumin gel in relation to plaque index and gingival index [74].

A study by Sha & Garib [78] investigated the antibacterial potential of turmeric gel against *P. gingivalis*, as well as connective tissue responses in the subcutaneous tissue of rats. The minimum inhibitory concentration (MIC) and the minimum bactericidal concentration (MBC) of curcumin against clinically isolated *P. gingivalis* were 12 μg/mL. Curcumin gel caused moderate inflammatory reactions at 7 and 30 days, while at 60 days, it caused a dramatic decline and led to an insignificant response [78]. In addition, curcumin gel stimulated rapid re-epithelialization, fibroblast proliferation, and healing by forming thick, well-organized bundles of collagen fibers. Curcumin has an effective antibacterial action against clinically isolated *P. gingivalis* at low concentrations (12 μg/mL) and has been considered a biocompatible material in subcutaneous tissues [78]. Rudhra et al. [79] investigated in vitro the antimicrobial efficacy of turmeric gel against *P. gingivalis* and *Prevotella intermedia*, using tetracycline gel as a positive control. The area of inhibition was measured in millimeters around both microorganisms. The maximum area of inhibition was obtained at 2% turmeric gel concentration, with a diameter of 10.3 mm compared to *P. gingivalis* and 11.4 mm compared to *P. intermedia* [79].

Nagasri et al. [16] investigated the effects of turmeric gel in combination with SRP on the presence of Socransky red complex periodontal pathogens, as assessed by polymerase chain reaction (PCR) tests; the application of the curcuma gel generated significant reductions in periodontal pathogens. Bhatia et al. [80], in a comparative interventional study in patients with severe periodontitis, administered 1% turmeric gel as an adjunct to initial SRP at one-, three-, and six-month intervals; the control group received SRP alone. At the end of the investigation period, they noticed significant reductions in probing depth, gingival bleeding, and periodontal clinical attachment loss in both groups, with more significant results in patients who also received adjunctive therapy with turmeric gel [80]. Moreover, this gel also significantly reduced the levels of *P. gingivalis*, *P. intermedia*, *Fusobacterium nucleatum*, and *Capnocytophaga* sp. [80]. In another study by Anitha et al. [81], the effects of applying 1% curcuma gel and 0.1% chlorhexidine gel as adjuvants to conventional nonsurgical therapy were compared; both groups showed significant reductions in clinical periodontal parameters, but also a statistically significant reduction in colony forming units; notably, these improvements were more important for the group of patients who underwent local applications of curcuma gel [81].

Sha et al. [82], in another study on rats with experimentally induced periodontitis, evaluated comparatively the effects of curcuma gel and chlorhexidine gel on serum RANKL and IL-1β concentrations; the authors concluded that turmeric gel generates chlorhexidine-like effects, regulating RANKL and IL-1β levels [82].

Chenar [83] evaluated the effect of curcumin gel, placed in periodontal pockets and isolated with Coe-Pak type periodontal cement for seven days, on serum levels of micronutrients (zinc, copper, and magnesium) and proinflammatory cytokines (IL-1β and TNF-α), as well as on probing depth, clinical attachment loss, plaque index, gingival index, and bleeding-on-probing index in patients with chronic periodontitis; curcumin gel had a significant effect on the reduction in clinical parameters but also in IL-1β, TNF-α, and copper (*p* ≤ 0.05), with an increase in zinc and magnesium levels after one month compared to the baseline value (*p* ≤ 0.05) [83].

Zambrano et al. [84] applied curcumin nanoparticles in periodontal pockets to assess the feasibility and biological effect of their local use; the authors noted that, after administration, inflammatory bone resorption was inhibited and, moreover, both the number of osteoclasts and inflammatory cells were significantly reduced. In contrast, in a study on the effects of local administration of 50 μg turmeric nanoparticles by Pérez-Pacheco et al. [85], statistically significant differences in periodontal parameters (probing depth, clinical attachment loss, and bleeding on probing) in IL-1α, IL-6, TNFα, and IL-10 molecules, and of a number of 40 bacterial species, evaluated at baseline, 3-days, and 15-days post-therapy, was not observed.

It also has been shown that the use of turmeric in slow degradation forms, such as films or chips, applied in the periodontal pocket, can generate beneficial effects in patients with periodontitis, materialized by improved clinical parameters [17,86,87].

## 5. Applications in Photodynamic Therapy

Photodynamic therapy (PDT) is based on the potential chemical effects of light, a method used in many forms of anticancer and anti-infective therapy [88]. Due to its beneficial properties, this form of therapy has been investigated in numerous studies in patients with periodontitis and peri-implantitis. Three main components are involved in photodynamic therapy: the light source, photosensitizer, and molecular oxygen. By combining these elements, a flow of reactions is generated, with therapeutic effect [89].

The use of PDT in periodontal adjunctive therapy involves the application of a photosensitizing substance in the periodontal pocket and its irradiation with a light source (laser or LED) of a length wave appropriate to the used substance [90]; cytotoxic reactive oxygen species are generated after exposure to light, with effects involving protein, cell membrane, and bacterial organ damage [91]. In addition, PDT has also demonstrated antiviral and antifungal effects, with the extension of the therapeutic potential in the oral cavity [92].

A photosensitizer or photoactivatable agent, such as methylene blue, is applied to the infected tissue. Exposure of tissue to light at the appropriate wavelength in the presence of molecular oxygen then generates the formation of reactive oxygen species (ROS) that cause nonthermal cytotoxic effects by damaging microorganisms’ proteins, cell membranes, and organelles [93].

Various substances have been investigated as potential photosensitizing agents; these include phenothiazine derivatives (methylene blue, toluidine blue), xanthene (erythrosine, eosin-Y, Bengal roses), riboflavin derivatives, indocyanine green, fullerene derivatives, and bordipyromethane derivatives [94]. Although most studies investigating PDT applications in periodontics involved the use of phenothiazine derivatives, attention has also been focused on the potential of curcumin as a photosensitizing agent. Curcumin absorbs light from the edge of UV and visible radiation, over a range of 300–500 nm, with the maximum absorption at about 420 nm [94].

It has been stated that the efficacy of a PDT protocol can be measured by the minimum concentration of photosensitizer that induces a 4-log decrease in survival for a given set of irradiation parameters [91]. Different turmeric-based photoactivation protocols have been investigated, demonstrating logarithmic reductions in different oral pathogens ranging from 7-log to 0-log [95].

Saitawee et al. [96] determined the antibacterial activity of various concentrations of photoactivated turmeric with LED light radiation at 420–480 nm wavelength for one minute against *A. actinomycetemcomitans* in an in vitro study. Bacteria with exponential growth were combined with a solution of curcuma concentration ranging from 25 to 0.098 μg/mL; 0.12% chlorhexidine solution was used as a positive control. In addition, the authors also evaluated the production of free radicals using electron spin resonance spectroscopy (ESR) with 5,5-dimethyl-1-pyrroline N-oxide (DMPO). The results demonstrated dose-dependent antibacterial activity of the turmeric solution [96]. In the absence of blue light irradiation, the turmeric concentration of 25 μg/mL resulted in a logarithmic reduction in *A. actinomycetemcomitans* of 6.03 ± 0.39 log10; turmeric concentration of 0.78 μg/mL under irradiation completely eradicated *A. actinomycetemcomitans*, an effect also obtained by co-culture with chlorhexidine 0.12%. At the same time, a maximum signal intensity of hydroxyl radical production was observed following the association of curcuma with LED irradiation [96].

Böcher et al. [92] evaluated the in vitro efficacy of an irradiated curcumin solution of 100 μg/mL using a 445 nm laser on *A. actinomycetemcomitans*, *Campylobacter rectus*, *Eikenella corrodens*, *F. nucleatum*, *P. gingivalis*, *P. intermedia*, *Parvimonas micra*, and *T. forsythia*. The authors failed to notice significant differences between the use of turmeric and the control solution, dimethylsulfoxide. The potential explanation for this inefficiency is that the wavelength was strictly 445 nm [92].

A study on a murine model with experimentally induced periodontitis [97] evaluated the influence of PDT with 40 μM turmeric solution and LED irradiation (468–485 nm) on alveolar bone loss in the furcation area in mandibular molars, compared with the application of turmeric without irradiation. The authors observed that PDT generated more favorable results related to decreased bone loss, a lower number of positive TRAP cells, a moderate immunolabeling for osteoprotegerin at 30 days, and a moderate to low RANKL immunolabeling pattern [97].

Sreedhar et al. [98] conducted a clinical and microbiological study on patients with periodontitis, divided into four groups: patients who underwent only scaling and root planing; patients who, in addition to SRP, also followed the single local application of curcumin gel for five minutes; patients with SRP + curcumin for five minutes + irradiation with blue light with a wavelength of 470 nm for five minutes on day “0”; patients with SRP + curcumin for five minutes + irradiation with blue light on days 0, 7, and 21. Clinical parameters (plaque index, probing depth, clinical attachment loss, and bleeding on probing), as well as microbiological parameters (*P. gingivalis*, *P. intermedia*, *A. actinomycetemcomitans*) were investigated at baseline, one month, and three months. The authors concluded that the additional use of curcumin gel generated an antibacterial effect on the periodontal pathogens investigated but PDT amplified the benefits of curcumin, enhanced even more by repeated sessions of PDT [98].

It was observed that the bactericidal effects of turmeric as a photosensitizing agent may depend on the type of carrier or solvent used, which may alter the wavelength at which the efficiency is maximum [99]. In addition, turmeric has a high rate of photodegradation, which translates into the need to use it almost immediately after preparation [100].

## 6. Immunological Considerations

Immunologically, curcumin inhibits TGF-β1-induced connective tissue growth factor expression by disrupting Smad2 signaling in human gingival fibroblasts [101].

Our need for knowledge focuses on the cellular signaling pathways involved in the development and proliferation of cancer, which are targeted by curcumin. Curcumin has been reported to modulate growth factors, enzymes, transcription factors, kinase, inflammatory cytokines, and proapoptotic (up-regulation) and antiapoptotic (down-regulation) proteins [102]. Periodontitis progresses due to elevated levels of active metalloproteinases (MMPs) and the imbalance between MMPs and their tissue inhibitors (TIMPs) [102]. Natural curcumin limits the lytic activity of MMPs but has low cellular absorption. The use of synthetic curcumin analogs exerted modulatory effects on the metalloproteinases’ activation, offering potential benefits in overcoming this limitation of treatment effectiveness [103].

To compare the effects of oral administration of natural curcumin and chemically modified curcumin (CMC2.24) on osteoclast-mediated bone resorption, apoptosis, and inflammation in a murine model of experimental periodontal disease, four groups were experimentally treated: the first with carboxymethylcellulose, the second with synthetic turmeric CMC2.24 30 mg/kg body weight, the third with natural turmeric in the curcumin dose of 100 mg/kg body weight, and the fourth group without treatment [103]. The results showed CMC2.24 and curcumin caused a significant reduction in cellular infiltrate inflammation; however, μCT analysis showed that only CMC2.24 reduced bone resorption and the number of TRAP-positive multinucleated cells (osteoclasts). Curcumin, but not CMC2.24, significantly reduced the number of apoptotic cells in gingival tissue and osteocytes in the alveolar bone crest [104].

*Curcuma longa* has various effects, such as antioxidant, antihyperlipidemic, antitumor, antimicrobial, anti-inflammatory, wound healing, and gastroprotective activities [105]. At the same time, however, it has been reported to cause contact dermatitis [106]. Similarly, kumkum, a powder based on turmeric and slaked lime, applied by Hindu women on the forehead, is known to cause allergic contact dermatitis. Possible contact allergens in kumkum include turmeric, Sudan-I, 4-aminoazobenzene, bright lacquer red R, and hemp oil [106]. To identify the allergen that causes allergic contact dermatitis induced by kumkum, patch testing is recommended with undiluted kumkum, undiluted turmeric, Sudan-1 (95%), 4-aminoazobenzene, and Indian standard series allergens. Several studies, including one by the North American Contact Dermatitis Group 2005–2006, have shown that allergic contact dermatitis to kumkum occurs due to both dyes (added for color enhancement) and turmeric [106]. All patients with suspected allergic contact dermatitis should be tested with kumkum, turmeric, and dye patches, on the basis of which a nonallergenic alternative material may be recommended [107,108].

Curcuma is nontoxic to humans, especially when administered orally [105]. Curcuma is also safe in animals. It is not mutagenic and safe during pregnancy in animals, but more human studies are needed. Oral use of turmeric was shown to be nontoxic to animal reproduction at certain doses [109]. Human studies showed no toxic effects and turmeric was safe at 6 g/day orally for 4–7 weeks. However, some side effects may occur, such as gastrointestinal disorders [109]. Among the anti-inflammatory effects, studies cite the protective effects of turmeric and its active ingredient, curcuma, against natural and chemical toxic agents [105].

Another point of view concerns the significant improvement in food allergy symptoms due to turmeric, in a model of food allergy in mice [110]. Turmeric as an antiallergic agent has shown immune-regulating effects by maintaining the Th1/Th2 immune balance, while curcuma has shown immunosuppressive effects. Mice were immunized with intra-peritoneal ovalbumin (OVA) and alum. Mice were orally challenged with 50 mg OVA and treated with turmeric extract (100 mg/kg) and curcumin (3 mg/kg or 30 mg/kg) for 16 days. Turmeric significantly alleviated the symptoms of food allergy (lower rectal temperature and anaphylactic response) induced by OVA, but curcumin showed a slight improvement. Turmeric also inhibited elevated OVA levels of IgE, IgG1, and mMCP-1. Turmeric reduced helper cell-related cytokines type 2 (Th2) and improved a Th1-linked cytokine [110].

Songkro et al. [111] conducted a study to evaluate the irritating effect of turmeric; three plants from the Zingiberaceae family (Ginger family) were selected, which are rich in essential oils, and the essential oils were extracted from the rhizomes by the method of hydrodistillation. These three plants were *Zingiber officinale* Roscoe, *Zingiber cassumunar* (Roxb), and *Turmeric zedoaria* (Berg) Roscoe, which are widely grown in Southeast Asia, including Thailand. The activity of improving the penetration of these three plants was investigated, using as a model Diclofenac sodium, a nonsteroidal anti-inflammatory agent, to evaluate the effect of these essential oils on skin permeability [111]. The potential for skin irritation and possible damage from the application of these essential oils to Wistar rat skin were further evaluated in vivo using histological examinations by light microscopy. The histopathological analysis revealed that the epidermis was thickened and had compact hyperkeratosis after the application of turmeric oil. The spinous cells were spongy and there was a moderate diffuse infiltration of inflammatory cells throughout the dermis, especially in the upper area. Focal areas of pustule formation were observed between epidermal cells. In conclusion, the effects on rat skin after exposure to 100% curcuma essential oil involved mainly the epidermis [111].

One last observation concerns the possibility of reducing the effect of curcuma by exposure to type B UV radiation. Plants are static and constantly exposed to environmental stress, including short-wavelength UV radiation. UV-B sensitivity varies between plant species as well as between different varieties and clones [112]. Some species are environmentally sensitive to UV-B, while others are highly tolerant of high UV-B (eUV-B) [113]. Constant exposure to eUV-B radiation develops a response to stress in plants through the regulatory feedback loop, such as activation of the UV Resistance Locus 8 (UVR8) pathway, which increases the production of anthocyanins and flavonoids contained in *Curcuma longa* [113].

## 7. Conclusions

In general, the results are in favor of using different forms of turmeric as an adjunct therapy to scaling and root planing. Nevertheless, it should be noted that most studies have a relatively small number of subjects included and, moreover, it seems that there is no consensus on the administration protocol (dosage, number of sessions). In addition, there is a need for long-term evaluation of the potential benefits of local administration of turmeric compounds.

Even if curcumin is generally nontoxic, the systemic status of the patient should be carefully assessed, in order to prevent a potential allergic reaction. Additionally, any risk of cumulative effect generated by turmeric-containing spice intake during local curcumin therapy should be considered.

## Figures and Tables

**Figure 1 pharmaceutics-14-00982-f001:**
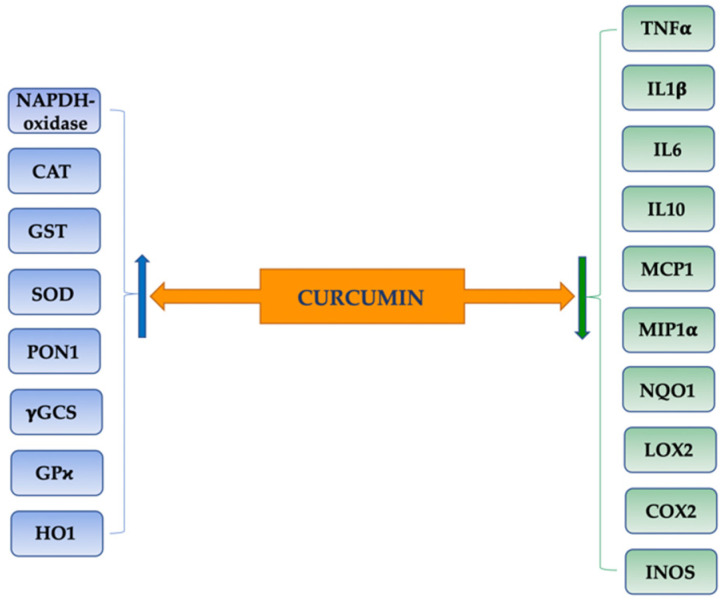
Molecular anti-inflammatory and antioxidant effects of curcumin. NAPDH: nicotinamide adenine dinucleotide phosphate; CAT: catalase; GST: glutathione S-transferase; SOD: superoxide dismuthase; PON1: paraoxonase-1; γGCS: γ-glutamylcysteine synthetase; GPχ: glutathione peroxidase-χ; HO1: heme oxygenase-1; TNFα: tumor necrosis factor-α; IL1β: interleukin-1β; IL6: interleukin-6; IL10: interleukin-10; MCP1: monocyte chemoattractant protein-1; MIP1α: macrophage inflammatory protein 1 α; NQO1: quinone oxidoreductase-1; LOX2: lipoxygenase-2; COX2: cyclooxygenase-2; INOS: nitric oxide synthase.

**Figure 2 pharmaceutics-14-00982-f002:**
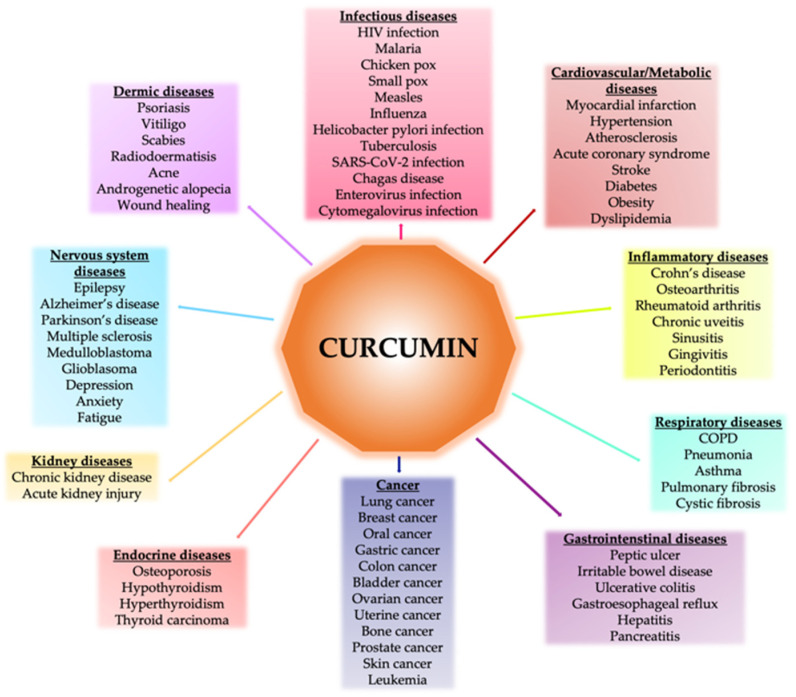
Effects of curcumin in different diseases.

## Data Availability

Not applicable.

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
