# Peer review of "Curcumin as a Natural Approach of Periodontal Adjunctive Treatment and Its Immunological Implications: A Narrative Review"

_pharmaceutics, 2022, doi:10.3390/pharmaceutics14050982_

Round 1

Reviewer 1 Report

Comments:

The topic of the present narrative review, discussing curcumin as a natural approach of periodontal adjunctive treatment and its immunological implications, seems interesting. The issue is novel

and has been adequately described.

Reviewed findings currently presented may pave the way for further clinical investigations and may be clinically relevant in the future in the perspective of periodontitis management both in health and illness.

The narrative review presented appears comprehensive and well structured, however Introduction and Conclusions sections needs to be expanded.

Editing for English language is needed.

Concerns and suggestions:

Title:

  • I would suggest to substitute the full stop with “:”.

Introduction:

  • Please, define periodontal disease (lines 32-33) as per Tonetti et al., (Tonetti, M.S.; Greenwell, H.; Kornman, K.S. Staging and grading of periodontitis: Framework and proposal of a new classification and case definition. J. Periodontol. 2018, 89, S159–S172) and briefly introduce the consequent jaw atrophy as per “Computed Tomography-Aided Descriptive Analysis Of Maxillary And Mandibular Atrophies. J Stomatol Oral Maxillofac Surg 2019;120(2):99-105. doi: 10.1016/j.jormas.2018.12.006”;
  • Please, modify microbial biofilm (line 33) with periodontal or gingival biofilm;
  • Please, substitute “disorganizing” with “disrupting” (line 41);
  • Please, substitute “with” with “recognizing” (line 43);
  • Please, remove “therapy” (line 46) because redundant;
  • Please, add to “antibiotics” and reference n. 8 Effectiveness of Antibiotics in Preventing Alveolitis After Erupted Tooth Extraction : A Retrospective Study. doi: 10.1111/odi.13297.

Paragraph 2:

  • Please prefer “demonstrated” to “demonstrate” (line 125);
  • Please, remove “local” (line 134).

Figure 2 legend:

  • Please, remove “forms of”.

Paragraph 3:

I would suggest to move period in lines 163-168 to line 138.

Conclusions section:

  • Expand on the putative inter-relation between periodontal disease and systemic inflammatory disorders (see Possible association of periodontal disease and macular degeneration doi: 10.3390/dj9010001 and Obesity and Periodontal Disease: a Narrative Review on Current Evidence and Putative Molecular Links. doi: 2174/1874210601913010526.), as well as benign and malignant neoplasms, especially emphasizing the need for the use of more effective adjuncts to periodontal treatment, both in active and maintenance phases;
  • Please, re-phrase period in lines 409-414;
  • Expand on your personal opinion and clinical recommendations.

Author Response

The topic of the present narrative review, discussing curcumin as a natural approach of periodontal adjunctive treatment and its immunological implications, seems interesting. The issue is novel and has been adequately described. Reviewed findings currently presented may pave the way for further clinical investigations and may be clinically relevant in the future in the perspective of periodontitis management both in health and illness.

Thank you for taking the time to analyse our manuscript and to give us the chance to improve it. Your suggestions have enriched our work and we have modified the paper accordingly.

  1. The narrative review presented appears comprehensive and well structured, however Introduction and Conclusions sections needs to be expanded.

Thank you for your excellent suggestion; we modified the manuscript, adding the recommended information.

  1. Editing for English language is needed.

English editing of the text was performed.

  1. Title: I would suggest to substitute the full stop with “:”.

Thank you for your valuable remark. The change was made in the title.

  1. Introduction: Please, define periodontal disease (lines 32-33) as per Tonetti et al., (Tonetti, M.S.; Greenwell, H.; Kornman, K.S. Staging and grading of periodontitis: Framework and proposal of a new classification and case definition. J. Periodontol. 2018, 89, S159–S172) and briefly introduce the consequent jaw atrophy as per “Computed Tomography-Aided Descriptive Analysis Of Maxillary And Mandibular Atrophies. J Stomatol Oral Maxillofac Surg 2019;120(2):99-105. doi: 10.1016/j.jormas.2018.12.006”.

Thank you for your suggestions; we added the proposed information.

  1. Please, modify microbial biofilm (line 33) with periodontal or gingival biofilm.

The term was changed.

  1. Please, substitute “disorganizing” with “disrupting” (line 41).

Thank you for your suggestion; we have changed the term.

  1. Please, substitute “with” with “recognizing” (line 43).

The change was made.

  1. Please, remove “therapy” (line 46) because redundant.

The term was removed.

  1. Please, add to “antibiotics” and reference n. 8 Effectiveness of Antibiotics in Preventing Alveolitis After Erupted Tooth Extraction: A Retrospective Study. doi: 10.1111/odi.13297.

Thank you for your suggestion; we modified the manuscript accordingly.

  1. Paragraph 2: Please prefer “demonstrated” to “demonstrate” (line 125).

Thank you for your input; we have changed the tense.

  1. Please, remove “local” (line 134).

Term was removed.

  1. Figure 2 legend: Please, remove “forms of”.

We changed the legend accordingly.

  1. Paragraph 3: I would suggest to move period in lines 163-168 to line 138.

Thank you for your valuable remark. The manuscript was modified accordingly.

  1. Conclusions section: Expand on the putative inter-relation between periodontal disease and systemic inflammatory disorders (see Possible association of periodontal disease and macular degeneration doi: 10.3390/dj9010001 and Obesity and Periodontal Disease: a Narrative Review on Current Evidence and Putative Molecular Links. doi: 2174/1874210601913010526.), as well as benign and malignant neoplasms, especially emphasizing the need for the use of more effective adjuncts to periodontal treatment, both in active and maintenance phases.

Thank you for your valuable remark; we have changed the text accordingly but we preferred to add the information in the Introduction section.

  1. Please, re-phrase period in lines 409-414.

Thank you for your suggestion; we have rephrased the whole Conclusions section.

  1. Expand on your personal opinion and clinical recommendations.

Thank you for your valuable suggestion. We have corrected the manuscript, adding the recommended information.

Reviewer 2 Report

1. The abstract describes the study well.
Keywords are well-chosen and reflect the study well.
3. Introduction: introduces the reader to the article's topic very well. It presents the theoretical information about the subject well.
4. All data presented by the authors are engaging, informative, and current. 
5. conclusions support the information presented by the authors. 
In the reviewer's opinion, the article can be published in its current form in MDPI.

Author Response

Thank you for taking the time to analyze our manuscript and thank you for your kind remarks.

Reviewer 3 Report

Dear authors, 

The study entitled "Curcumin as a Natural Approach of Periodontal Adjunctive Treatment and Its Immunological Implications. A Narrative Review." is a well-written manuscript. However, the manuscript looks more like a book chapter than a scientific article.

There is no information about Curcumin in the Introduction. Only in item 2. 

The authors should add this info in the introduction. 

Everyone knows that a narrative review usually doesn’t present the search methods. However, it reduces the study's quality. 

The authors should review it and organize the search methods. After that, you will have a scoping review. After that, the text will sound more scientifically than now. 

Organize a table with the relevant information from the articles.

Prepare a flowchart of your screening method.

I miss the Discussion of your manuscript.

Please, include the limitations of your research.

The conclusion doesn't conclude anything. 

Also, the last paragraph doesn’t make any sense ("Moreover, the allergist's advice would be to use turmeric with jurisprudence in all cases of periodontitis ...")

The Dentist has the autonomy to prescribe any medication which will help the patient's oral health. 

Keep it in mind.

In the end, the manuscript is informative. However, it doesn’t sound like a scientific paper.

Regards,

#Reviewer#

Author Response

The study entitled "Curcumin as a Natural Approach of Periodontal Adjunctive Treatment and Its Immunological Implications. A Narrative Review." is a well-written manuscript. However, the manuscript looks more like a book chapter than a scientific article.

  1. There is no information about Curcumin in the Introduction. Only in item 2. The authors should add this info in the introduction. 

Thank you for your input. We consider that the whole manuscript focuses on curcumin, in its various forms as periodontal adjunctive therapy and our way of writing the Introduction paves the pathway for all the subsequent Sections.

  1. Everyone knows that a narrative review usually doesn’t present the search methods. However, it reduces the study's quality. The authors should review it and organize the search methods. After that, you will have a scoping review. After that, the text will sound more scientifically than now. Organize a table with the relevant information from the articles. Prepare a flowchart of your screening method. I miss the Discussion of your manuscript. Please, include the limitations of your research.

We want to thank you for your suggestions. Nevertheless, our manuscript was written as a Narrative Review and we would like to maintain its original intention. All your remarks are very correct and appliable to a Scoping Review but, with all due respect to the honorable Reviewer, we did not intend to write a Scoping Review.

  1. The conclusion doesn't conclude anything. Also, the last paragraph doesn’t make any sense ("Moreover, the allergist's advice would be to use turmeric with jurisprudence in all cases of periodontitis ..."). The Dentist has the autonomy to prescribe any medication which will help the patient's oral health. Keep it in mind.

Thank you for your remark. We have rephrased the Conclusions section. All the authors of the manuscript are specialists with a considerable number of years of expertise and, as such, we still consider that, even if a dentist has the legal means to recommend a certain treatment, the patient’s exposure to any type of drug should be based on common medical sense, with all the potential systemic factors and risks taken into account.

  1. In the end, the manuscript is informative. However, it doesn’t sound like a scientific paper.

We want to thank you for taking the time to analyse our manuscript.

Reviewer 4 Report

In this review, the authors go over the periodontal therapy and the problems therein. The effect of curcumin is clearly described based on the previous studies. Overall, this manuscript is very informative. Some minor revision is suggested.

  1. The scientific names of species should be italicized. “Curcuma longa” should be italicized.
  2. It is better to enlarge Figure 2 for the ease of reading.
  3. The title of reference 16 seems to be miss-typed.

Author Response

In this review, the authors go over the periodontal therapy and the problems therein. The effect of curcumin is clearly described based on the previous studies. Overall, this manuscript is very informative. Some minor revision is suggested.

  1. The scientific names of species should be italicized. “Curcuma longa” should be italicized.

Thank you for taking the time to analyse our study and to give us the chance to improve it. We apologize for this mistake on our behalf. We have corrected the manuscript accordingly.

  1. It is better to enlarge Figure 2 for the ease of reading.

Thank you for your remark; we enlarged Figure 2.

  1. The title of reference 16 seems to be miss-typed.

Thank you for pointing this out. The whole References Section was revised and corrected.

Round 2

Reviewer 1 Report

Comments:

Suggestions for the Authors have been welcomed and the manuscript has been improved.

Further minor suggestions for the authors are detailed below.

Suggestions

Introduction:

Please substitute “which help on the maintenance and” with “supporting and maintaining the” (lines 32-33)

Please, eliminate often (line 34)

Please, add to “, to which the host will react through non-specific and specific defense 35 systems, generating a cascade of inflammatory reactions, in which oxidative stress also 36 plays an important role [3]” “, which sustains the disease” (lines 35-37);

Please, substitute “begins with epithelial attachment loss, clinically detectable by periodontal probing [5], and it evolves to deeper tissues, with bone destruction” with “proceeds  in apical directions [5], involving deeper tissues and causing bone destruction” (lines 40-41).

Author Response

Suggestions for the Authors have been welcomed and the manuscript has been improved.

We want to thank you for taking the time to analyze our manuscript and to give us the chance to improve it. We have modified the paper in accordance to your suggestions.

Further minor suggestions for the authors are detailed below.

Introduction:

1.Please substitute “which help on the maintenance and” with “supporting and maintaining the” (lines 32-33)

Thank you for your suggestion; the substitution was done.

2.Please, eliminate often (line 34).

The word was removed.

3.Please, add to “, to which the host will react through non-specific and specific defense 35 systems, generating a cascade of inflammatory reactions, in which oxidative stress also 36 plays an important role [3]” “, which sustains the disease” (lines 35-37).

Thank you for your input; we have added the recommended sentence.

4.Please, substitute “begins with epithelial attachment loss, clinically detectable by periodontal probing [5], and it evolves to deeper tissues, with bone destruction” with “proceeds in apical directions [5], involving deeper tissues and causing bone destruction” (lines 40-41).

Thank you for your valuable remark; we have changed the phrase accordingly.

Reviewer 3 Report

The authors provided all the modifications suggested by this reviewer.

Author Response

We want to thank you for taking the time to analyze our manuscript and for all your efforts.